# Development of New Emission Reallocation Method for Industrial Source in China

Yun Fat LAM[1], Chi Chiu CHEUNG[2], Xuguo ZHANG[3], Joshua S. FU[4], Jimmy Chi Hung FUNG[3]

[1]Department of Geography, University of Hong Kong, HKSAR, China
[2]ClusterTech Limited, HKSAR, China
[3]Institute for the Environment, Hong Kong University of Science and Technology, HKSAR, China
[4]Department of Civil and Environmental Engineering, University of Tennessee, Knoxville, USA

*Correspondence to*: Yun Fat LAM (yunlam@hku.hk)

**Abstract.** An accurate emission inventory is a crucial part of air pollution management and is essential for air quality modelling. One source in an emission inventory, an industrial source, has been known with high uncertainty in both location and magnitude in China. In this study, a new reallocation method based on blue-roof industrial buildings was developed to replace the conventional method of using population density for the Chinese emission development. The new method utilized the zoom level 14 satellite imagery (i.e., Google®) and processed it with Hue, Saturation, Value (HSV)-based colour classification to derive new spatial surrogates for province-level reallocation, providing more realistic spatial patterns of industrial $PM_{2.5}$ and $NO_2$ emissions in China. The WRF-CMAQ based PATH-2016 model system was then applied with the new processed industrial emission input in the MIX inventory to simulate air quality in the Greater Bay Area (GBA) area (formerly called Pearl River Delta (PRD)). In the study, significant Root Mean Square Error (RMSE) improvement was observed in both summer and winter scenarios in 2015 when compared with the population-based approach. The average RMSE reductions (i.e., 75 stations) of $PM_{2.5}$ and $NO_2$ were found to be 11 $\mu g/m^3$ and 3 ppb, respectively. Although the new method for allocating industrial sources didn't perform as good as the point and area based industrial emissions obtained from the local bottom-up dataset, it still showed a large improvement over the existing population-based method. In conclusion, this research demonstrates that the blue-roof industrial allocation method can effectively identify scattered industrial sources in China and is capable of downscaling the industrial emissions from regional to local levels (i.e., 27 km to 3 km resolution), overcoming the technical hurdle of ~10 km resolution from the top-down or bottom-up emission approach under the unified framework of emission calculation.

Keywords: Industrial Emissions, MIX Emission Inventory, Emission Surrogate, Satellite Image Processing, and $PM_{2.5}$ air quality

# 1 Introduction

The emission inventory is essential for air quality management and climate studies. Various applications, including setting up regional emission reduction target and performing numerical air quality forecasts, rely upon an accurate inventory for sound assessment and judgment (Krzyzanowski, 2009; Zhao et al., 2015). As the purpose and type of emission inventories (e.g., point, mobile and area) vary largely, data requirement and collection method can be quite different (Kurokawa et al., 2013). In point source inventory, collecting large point sources (i.e., power plant) is generally straightforward, while obtaining

data from scattered industrial sources often poses challenges and requires tremendous effort to collect and process. In developed countries like the USA, industrial sources are usually large, but yet it does not always contribute to a dominant portion in the emission inventory (e.g., 10-15% in $PM_{10}$ and 25-60% in NMVOC), and its data collection process is commonly incorporated into routine permitting exercise, making it easy to be included in their national inventory (ECCC, 2017; Janssens-Maenhout et al., 2015; Lam et al., 2004). Unfortunately, this is not the case for developing countries like China where industrial

sources are considered as a major emitter; Li et al. (2017) reported that industrial sources from non-power generation are the largest contributors of $PM_{10}$ and NMVOC in the MIX inventory. With the infinite numbers of small factories scattered across the continent with frequent change of location caused by urban redevelopment, these industrial sources are often treated as area/nonpoint source regardless of whether it is a point source (e.g., stack) or not. Hence, it possesses large spatial uncertainties in the inventory.

In recent years, various Asian emission inventories (e.g., REAS, MIX, and MICS-ASIA) have been developed for the purpose of air pollution modelling, and it has been widely used for studying transboundary air pollution among the Asian countries (Chen et al., 2019; Tan et al., 2018). The top-down or semi bottom-up approach based on the unified framework of source categories, calculating method, chemical speciation scheme, and spatial and temporal allocations was commonly used in the emission inventory development, where emissions were handled separately for different source categories (e.g., power,

industry, transport, residential/domestic and agriculture) with limited spatial resolutions ranged from 10 to 27 km (Kurokawa et al., 2013; Li et al., 2017; Ohara et al., 2007; Streets et al., 2003). In some cases, higher resolution emission inputs were achieved via GIS spatial interpolation for subregional (i.e., 10-15 km resolution) application. Information such as stack location, road network and population density was applied as surrogate data for spatial reallocation (Du, 2008). For the case when ultra-high resolution (i.e., 1-3 km resolution) was needed, it was often supplemented with the bottom-up approach using the detailed

activity data (i.e., exact emission locations and its relevant emission amounts) in the emission inventory development (HKEPD, 2011). For the category of small/medium industrial sources where location information was frequently missing in the top-down/semi bottom-up emission approach, the population density was used as the surrogate, giving the fact that population density was considered a good proxy for accessing employment, goods and services (Giuliano and Small, 1993). Historically, this approach seemed to be quite robust to capture the factory location due to: 1) the Danwei/socialist work units which

enforced jobs and residences to be closed to each other to reduce travel distance (Yang, 2006), and 2) factory jobs were typically included accommodation (i.e., dormitory) for attracting foreign workers. With limited transport infrastructure, dormitories were usually within a few kilometres away from the factories. However, in recent years, the land-use and housing reforms in China has led to a spatial separation of jobs and residences, the strong factory-residence pattern has slowly diminished in Chinese cities as efficient public transportation has emerged. With the adaptation of industrial park in the urban

renewal process, it further expedited the separation of industrial-related employment (in the outskirt of the city) from residential space (city centre) (Zhao et al., 2017). As a result, there is a need to reconsider how industrial emissions are handled in the top-down/semi bottom-up emission approach, searching for a suitable surrogate for the emission reallocation.

        In this study, the concept of a blue-roof industrial surrogate was introduced for the first time for Chinese industrial emission allocation. The approach assumed the majority of industrial buildings (both factories and its warehouses) in China

were single-story non-concrete buildings with their rooftop was made out of galvanized metal coated with blue epoxy. In this

development, satellite imagery with zoom level 14 (i.e., Google®) was adopted and processed with HSV-based colour classification for generating the province-level spatial surrogate. The Community Multi-Scale Air Quality (CMAQ) based PATH-2016 platform with 3km MIX inventory was then applied to evaluate the impacts of air quality predictions between population and blue-roof based methods, and the simulated results of $PM_{2.5}$, $NO_2$, and $O_3$ were then compared to local observation data and CMAQ results from the point/area based bottom-up approach (hereafter referred to as "btmUp case") from Zhang et al. (2020) to assess its model performance (HKEPD, 2011; Li et al., 2017).

## 2    Methodology

A new allocation method called "blue-roof" industrial allocation was introduced in the top-down/semi bottom-up emission approach for better allocating the scattered Non-Power Generation (NPG) industrial emissions in China. In this study, the CMAQ model and the regional Asian emission inventory-MIX were applied to evaluate the effectiveness of the new allocation method on the performance of air quality prediction. The target simulation year is 2015, and the details of each component are described below:

### 2.1    Study area, simulation domain, and observation network

The Guangdong-Hong Kong-Macao Greater Bay Area (GBA), also known as the Pearl River Delta (PRD), was adopted as the study area for the industrial allocation test. The characteristic of diverse industrial clusters (e.g., garment, electronics, and plastic factories) scattered across the area creates an ideal testbed for spatial examination. The GBA area consists of two special administrative regions (Hong Kong and Macao) and nine Chinese municipalities, including Guangzhou, Shenzhen, Zhuhai, Foshan, Huizhou, Dongguan, Zhongshan, Jiangmen, and Zhaoqing in Guangdong Province with a total area coverage over 56,000 $km^2$. It is classified as one of the world-class manufacturing hubs in China. In this study, the CMAQ based PATH-2016 was adopted to evaluate the influence of air quality prediction from the new allocation method. The PATH-2016 modelling platform consists of 4 nested domains, including East Asia and Southeast Asia (D1), Southeastern China (D2), and GBA)/PRD (D3), and Hong Kong (D4) with resolutions of 27 km, 9 km, 3 km and 1 km, respectively. For this study, only D1-D3 was applied as it already covered the entire GBA with a reasonable spatial resolution (e.g., 3 km) for regional air quality simulation, as shown in Figure 1. Details of PATH-2016 and its model setting are discussed in the later section. For evaluating the performance of CMAQ air quality prediction, the China National Environmental Monitoring Centre (CNEMC) and the Guangdong-Hong Kong-Macao Pearl River Delta Regional Air Quality Monitoring Network (HKEPD, 2016) with over 75 surface observation stations were adopted (available at http://www.cnemc.cn/, last access: 10 September 2020). These stations measure various air pollutants, including $PM_{2.5}$, $PM_{10}$, $SO_2$, $NO_2$, and $O_3$.

### 2.2    PATH-2016 and MIX inventory

The PATH-2016 is a WRF-CMAQ (Community Multi-Scale Air Quality model) Air Quality system used by the HKSAR government for air quality-related policy. It has been validated in several studies (HKEPD, 2011, 2019; Zhang, 2020). In this study, CMAQ version 5.0.2 with AERO5 aerosol module and CB05CL carbon bond chemical mechanism driven by WRF version 3.7.1. was adopted for air quality simulation. The model setup for CMAQ simulation is summarized in Table 1, and WRF meteorological validation can be found in Zhang (2020) and HKEPD (2019). The initial/boundary conditions for the outermost domain, D1 was generated from the global model GEOS-Chem outputs for Asian pollution background (Lam and Fu, 2010). In terms of the model emissions, the majority of the anthropogenic emissions were adopted from the Asian emission inventory, MIX. The MIX is a regional emission inventory developed to support the Model Inter-Comparison Study for Asia (MICS-Asia) and the Task Force on Hemispheric Transport of Air Pollution (TF HTAP) (Li et al., 2017). It consists

of 5 anthropogenic emission source categories, including point, industry, transport, residential, and agriculture (NH$_3$ only) with a resolution of 0.25° (~27km), and its emission base year is for 2010. In this study, the MIX inventory was first scaled to the target simulation year of 2015 based on available sector-based control technologies (Li et al., 2019; Zhang, 2020; Zheng et al., 2018). The derived emission totals from each sector, except for the industrial emissions, were then temporally and spatially interpreted into 27km (D1), 9km (D2), and 3km (D3) resolutions using the top-down emission method described in Du (2008). Detailed methodology and validation of the base year 2015 emission inventory were extensively discussed and can be found in our previous publications (Zhang et al., 2021; Zhang et al., 2020). As Hong Kong emissions were not well presented in the MIX inventory due to the limitation of spatial resolution, the bottom-up emissions from the PATH-2016 platform were adopted for Hong Kong emissions. For the remaining sectors that were not available from the MIX inventory, it was adopted from the PATH-2016 study (Zhang, 2020). These include MEGAN biogenic, GFED biomass burning, Marine and sea-salt emissions (Athanasopoulou et al., 2008; Giglio et al., 2013; HKEPD, 2019; Ng et al., 2012).

## 2.3    Case study for nonpoint source industrial allocation

The purpose of CMAQ simulation is to evaluate the performance of the new industrial allocation method on its effect on air quality prediction in China. Two CMAQ scenarios tailored for the industrial allocation methods were proposed and tested with the MIX inventory. These scenarios are 1) population-based method and 2) blue-roof based method. Details of each method are described in the later section. For each scenario, two months of CMAQ air quality simulation were performed. The selection of winter (January) and summer (August) months from 2015 was to allow better reflection of air quality impacts from the change of Asian monsoon in Southern China. The choice of using 2015 as the based year was to permit more local observation data to be available in the GBA area from the Chinese national observation network (operated after late 2013), moreover, to better fit with the modelling effort in 2016 Air Quality Objectives (AQOs) review that has also applied PATH-2016 model (Zhang, 2020).

### 2.3.1    Base case - population-based method

The population-based method (hereafter referred to as "base case") is commonly applied in the top-down emission inventory to allocate residential (area) or NPG industrial sector in the regional emission inventory (Du, 2008; Li et al., 2017; Ohara et al., 2007). It utilizes population or population density as a spatial proxy to distribute the sectorial emissions into the simulation grids. It assumes a strong association is present between population and industrial emissions. In this study, the Oak Ridge National Laboratory (ORNL)'s LandScan global population data with the resolution of ~1 km (30″ X 30″) gridded spatial resolution was applied to allocate industrial emissions. To allow the separation of urban population from the rural population in LandScan grids, a threshold value of 1,500 people per square kilometres was adopted, in which any grid values that were greater than this number were considered as urban grids (Liu et al., 2003). The basic equation for estimating the gridded emissions ($E_{m,n}$) using urban population is shown in Eq (1). $E_m$ is the total emission in a province ($m$), and $U\_pop_n / U\_pop_m$ is the ratio of urban population from a grid ($n$) to the province total. This value is commonly referred to as the spatial allocation factor, and it is a dimensionless value ranged between 0 to 1. The collection of spatial allocation factors in the gridded matrix is called a spatial surrogate. In this study, all the calculations and the spatial interpretation were performed in ArcGIS to yield CMAQ-ready emissions.

$$E_{m,n} = E_m \times \frac{U\_pop_n}{U\_pop_m},\qquad\qquad\qquad(1)$$

where $E_{m,n}$ is the emission in $n^{th}$ grid for $m^{th}$ province, $E_m$ is the total emission in $m^{th}$ province, $U\_pop_n$ is the urban population count in $n^{th}$ grid, $U\_pop_m$ is the total population in $m^{th}$ province.

### 2.3.2    Blue-roof case: blue-roof based method

The "blue-roof based method" (hereafter referred to as "blue-roof case") adopted the concept of rooftop colour for associating the location of industrial buildings. In China, warehouse and industrial rooftops are commonly made out of galvanized metals with a coat of colour epoxy (i.e., light blue, green, pink, and purple). As more than 90% (by general observation) of these industrial roofs are in light blue, this unique feature was captured and applied to develop the new allocation method. To derive the allocation factor for each grid for the modelling domain, satellite imagery was used. Among different imagery products (e.g., Google®, Bing®, Baidu®, Latsat8, and SPOT7), the Google imagery was selected for the basis of the analysis, as it provided their products with less processing effort for different zoom levels. Overall, the zoom level 14 or above (~ 9.5 metre/pixel or above) has been confirmed to be sufficient for use in the colour detection process for major industrial rooftops in China. Considering the study required to process a relatively large area, the lowest possible zoom level (i.e., 14) was preferred. In this development, the QGIS platform v2.16 with two plugins (i.e., OpenLayers and Python) was adopted to provide a smooth process of overlaying satellite imagery with the Google Maps API v3 and to perform a pixel-based HSV colour detection (i.e., OpenCV) in QGIS. The colour detection method utilized HSV colour space for better identifying blue colours in given images. The choice of using OpenLayers (OL) at that time was to avoid the violation of the Terms Of Service (TOS) regarding the direct usage of Tile Map Services. As for now, this type of operation is no longer allowed under the updated TOS by Google®. To perform a similar process, one may choose to use Google Earth Engine or Bing/Baidu Maps API with OL. Figure 2 shows the basic flow chart of the process. Overall, about 2,000 map tiles that contained "blue-roof" were processed using the HSV algorithm for the area of the D3 domain. At last, the identified blue-roofs were then converted into polygons and stored into a shapefile. As the HSV algorithm was unable to distinguish the blue ocean and river features from blue-roofs, a removal process using the shapefiles of coastal line and inland waterbody was applied to eliminate the falsely identified waterbody using ArcGIS. The resulted blue-roof shapefile was then spatially interpreted with China administrative (i.e., province) boundaries and CMAQ raster grids to yield the information of gridded blue-roof areas ($B\_area_n$) and total blue-roof areas for each province ($B\_area_m$). This information was further applied to province total emission ($E_m$) to calculate the gridded emissions ($E_{m,n}$) using Eq (2):

$$E_{m,n} = E_m \times \frac{B\_area_n}{B\_area_m},  \tag{2}$$

where $E_{m,n}$ is the emission in $n^{th}$ grid for $m^{th}$ province; $E_m$ is the total emission in $m^{th}$ province; $B\_area_n$ is the total blue-roof area in $n^{th}$ grid; $B\_area_m$ is the total blue-roof area in $m^{th}$ province.

## 3    Results and discussion

### 3.1    HSV value selection, data training, and results of blue-roof colour identification

The satellite imagery of Google Map Tiles with zoom level 14 (which was retrieved between 2015-16 for this study) had exhibited a colour variation due to the inconsistent environmental conditions (e.g., cloud cover, visibility, and brightness of the day) when the images were taken. As these collective images were taken from different seasons or years, the aggregated images might not reflect a single snapshot of a specific time. To determine suitable parameters for the HSV algorithm, an optimization process that iteratively searches for high hit rates, low false detection and false alarm rates (See Supplement Eq(S1-S3) for definition) was applied. Three urban areas are Jing-Jin-Ji (Baoding area with 332 km$^2$), Yangtze River Delta (Shanghai area with 1,336 km$^2$), and GBA (Fushan area with 1,194 km$^2$) were picked as the training dataset as we recognized that cities and regions might have their own building styles and development patterns, choosing these three regions not only allowed more diverse samples to be included in the training dataset but also incorporated the potential effect of solar incident

angles on image colour (i.e., different brightness) under different latitudinal positions and time of satellite passing. To obtain the "ground truth" reference for iterative comparison, manual digitization of blue-roofs using the zoom level 16 data was performed for those three areas. The result of the iterative process shows that not a single set of HSV ranges was sufficient to capture the blue colour variation exhibited in the google images. As there was a broad spectrum of blue colours (e.g., low cyan, cyan blue, low blue) found in the satellite images, four sets of HSV ranges were used for the blue roof identification algorithm,

in which each set of HSV ranges were adopted to identify an independent section of "blue colour" from the HSV solid cylinder. It should be noted that as the ranges of HSV values are considered as business confidential information under the project agreement, the exact values are not disclosed here. In general, the applied HSV values were ranged between 193° and 230° for Hue (H), 17% and 90% for Saturation (S), and 40% and 100% for Value (V). Figure 3a-c shows samples of training images (100 km$^2$) in Baoding, Shanghai, and GBA, and Table 2 shows the summary of the training performance.

Overall, 74% to 88% (hit rates) of the blue roof areas were successfully identified by the algorithm, while the false detection rates and false alarm rates were ranged between 35% to 51% and 0.1% to 0.5%, respectively. The low percentages of false alarm rates indicate only a small amount of non-blue-roof areas were included by the algorithm. For a closer look at the results of false detection rates, it reveals that the false identification was mainly concentrated around the building boundaries because of the fuzziness of the building edges from the zoom level 14 satellite images. The percentages of false

detection rates varied for images in different areas as the clearness of satellite images depending on the atmospheric conditions (e.g. cloudiness, air pollution, etc.) at the time they were taken. Comparing with the images of Shanghai and GBA, the Baoding image has a relatively higher degree of blurriness which explained why the false detection rate for the Baoding was higher than those in the other two training areas. While the false selection around the building edges may incur different levels of errors to the blue-roof identification result, however, it does not generally affect the spatial distribution of the blue roof areas

selected by the algorithm. It is observed that the GBA area has a low hit rate (i.e., 74%) due to more scattered/isolated development than the other areas. In Boading and Shanghai, industrial parks are more common than in GBA. Buildings clustered together might have contributed to higher hit rates, but at the same time caused high false detection rates, as the gap between buildings was also included as blue-roof. To better evaluate the algorithm response to different environmental condition, another 7 areas (approximately 100 km$^2$ each) covering a wide variety of geographical locations and features were

selected to validate the blue-roof identification algorithm, as shown in Figure 3d-j. The results in Table 2 show that the algorithm achieved between 76-92%, 9-54%, and 0-0.3% for the hit rates, false detection and false alarm rates, respectively. These similar results found in Nagqu, Baoji, Kaifeng, Xi'an, and Zhengzhou indicate that the algorithm is relatively stable across the continent of China. For the remote areas in Taklimakan Desert and Yunnan (Figure 3i and 3j), it should be noted that the "ground truth" blue-roof areas were zero, so the hit rates were inapplicable for these two test images.

**3.2 Blue-roof allocation process and CMAQ ready emission**

   Large spatial data with various geospatial information (e.g., province shapefile) was processed through ArcGIS to create the gridded emissions for the CMAQ model. As mentioned, the selected blue-roof areas from the HSValgorithm have first undergone a spatial operation for removing the falsely identified water bodies from the blue-roof dataset. The data was then used to compute the gridded total blue-roof areas in each grid (See Figure 4a). Furthermore, the total blue-roof areas in each

220 province (i.e., Guangdong, Guangxi, and Jiangxi) within the domain were also generated. Figure 4b shows the province spatial surrogate (i.e., $\frac{B\_area_n}{B\_area_m}$ in eq (2)) that was created for the GBA emission allocation. The values in each grid in the surrogate file should fall between 0 to 1, and the total in the province should sum up to 1.0 for data integrity. The yellow grids (values greater than 0) were clustered around the centre of PRD, reflecting the high density of blue-roof buildings were identified along the pearl rivers, and the yellow lines extended from PRD indicates that small remoted industrial areas were built along

the major highways in GBA. For the grids with magenta (values with 0), it is confirmed to be hilly areas, forests, or water reservoirs, no blue-roof building was identified.

At last, for generating the final gridded industrial emissions, the spatial surrogate was applied to the MIX industrial emissions to spatially allocate the province-level emissions into the CMAQ grids. All sectoral emissions, including power, transportation, industrial, residential, agriculture and others were aggregated together with the newly produced industrial emissions to generate the CMAQ ready gridded emissions for air quality simulation. Figure 5 shows the daily column total of CMAQ ready $PM_{2.5}$ emissions (January 1, 2015) for the base case, blue-roof case, and point/area based btmUp case from Zhang et al. (2020). In general, more spatial spreading is observed in the blue-roof case than in the base case within the GBA area, but the spread is not as wide as in the point/area based btmUp case. The widespread of $PM_{2.5}$ emission in the btmUp case is attributed to the inclusion of both industrial point and industrial area sources, which was not applied the same way as in the base and blue-roof cases. In the base case (Figure 5a), emission (over 200 g/s/grid of $PM_{2.5}$) is intensely clustered around the city centres of Guangzhou (GZ) and Foshan (FZ). A circular belt of intense $PM_{2.5}$ emission is observed along the coast of Zhujiang River Estuary in Shenzhen (SZ). In contrast, $PM_{2.5}$ emission in the blue-roof case (Figure 5b) is more widely spread across the region, with additional focuses in Dongguan (DG) and north of Zhongshan (ZS). The hotspots of $PM_{2.5}$ exhibited in (Figure 5b) are strongly aligned with the spatial pattern of hotspots from Cui et al. (2015), which was generated from the source apportionment method. In Shenzhen, the circular belt of high emission previously observed in the base case has been disappeared. Further investigation shows that the coastal area along the Zhujiang River Estuary has already been converted into recreation and residential areas. Due to the high population density found in the area, the base case had allocated a large amount of $PM_{2.5}$ emission to the area. For small industrial areas, the blue-roof case also seems to outperform the base case as it has identified more scattered industrial areas in the region. As shown in BE (Figure 5), the blue-roof method is capable of capturing the small industrial towns (See supplementary Figure S1) along the major highways. For this particular example, the industrial area captured by the blue-roof method is located in the rural area of Qingyuan with over 20 petrochemical factories or warehouses, which has been missed in the base case. When comparing the blue-roof case with the point/area based btmUp case (Figure 5c), clear spots of $PM_{2.5}$ underestimation were observed which are shown in the square boxes of Figure 5b pointing at the northeastern and southwestern sides of PRD, and north of Guangzhou. As the focus of the study is to investigate the improvement of the blue-roof surrogate in the MIX industrial sector, rather than the performance differences between the MIX unified emissions and local bottom-up emissions. Therefore, instead of showing the uncertainty of emission inventory which is infeasible here, we have developed spatial blue-roof surrogate (Figure 4b), the comparison of the model-ready emissions (Figure 5), and the time series plots of typical stations (Figure 7) to illustrate the performance of the blue-roof algorithm.

### 3.3 CMAQ simulated air quality and statistical comparison

### 3.3.1 Performance comparison between the base case and blue-roof case

The CMAQ simulation was performed on both base case and blue-roof case to evaluate the air quality impacts of using different allocation methods for industrial emissions. In addition, to better understand how good the blue-roof method performs, the CMAQ results using the local point/area based btmUp emission method adopted from Zhang et al. (2020) were also included in the comparison. Figure 6 shows the simulated monthly average surface $PM_{2.5}$ for base case (a, d), blue-roof case (b, e), and point/area based btmUp case (c, f); the left (a-c) and right (d-f) panels represent the January and August cases, respectively. As expected, the base case (top panel) has much lower spatial spreading when comparing with the blue-roof (middle panel) and the point/area based btmUp (bottom panel) cases illustrated in the earlier section. The wider spreading of $PM_{2.5}$ in the blue-roof case (middle panel) was attributed to the redistribution of industrial emissions from highly populated areas (i.e., Guangzhou, Foshan and Shenzhen) found in the base case into other areas of GBA. The redistribution process has lowered the CMAQ prediction for those three urban areas, moreover, to reduce the $PM_{2.5}$ prediction along the coast of Zhujiang

River Estuary at the circular belt of Shenzhen, which was mentioned in Figure 5a). As monsoon wind runs differently in summer and winter, it affects the regional pollutant transport and air quality prediction in the GBA area. For the winter case, with the effect of the northeast prevailing wind in January, the reduction of Guangzhou and Foshan emission/pollution (See Supplement Fig. S2) has a strong positive impact on both local and downwind regions (i.e., south of Guangzhou - Zhongshan, Zhuhai, and Macau). This is illustrated by the $PM_{2.5}$ time-series plot of Zhongshan station (22°31'16.0"N 113°22'36.8"E) shown in Figure 7a) where the large $PM_{2.5}$ overestimation in the base case (grey line) was significantly reduced into the more acceptable range shown in the blue-roof case (blue line). The Root Mean Square Error (RMSE) was trimmed down nearly in half by about 23.0 $\mu g/m^3$ (from 49.2 $\mu g/m^3$ in the base case to 26.2 $\mu g/m^3$ in blue-roof case), demonstrating the effectiveness of using the blue-roof allocation method in the top-down emission approach. This is not entirely the same case for summer (August) when the prevailing wind is from the southwest that brings clean marine boundary to the region. Although, in some situations, due to the presence of distant typhoon (e.g., Soudelor (August 4-11) and Goni (August 21-25)), the outermost of typhoon circulation had forced the wind direction changed to northeasterly and resulted in a similar transport phenomenon that causes the $PM_{2.5}$ spikes in summer (Lam et al., 2018). For the general situation during the non-typhoon condition, stronger $PM_{2.5}$ underestimation is observed in the blue-roof case than in the base case, worsening the Mean Bias (MB) from -7.0 in the base case to -12.4 $\mu g/m^3$ in the blue-roof case. In terms of RMSE, there is nearly no difference between the base case (i.e., 20.47 $\mu g/m^3$) and blue-roof case (i.e., 20.48 $\mu g/m^3$), as the performance degradation observed during the non-typhoon period was compensated by the improvement during the typhoon period. Figure 8 shows the comparison of spatial performance between the base and blue-roof cases. The "RMSE improvement" means that the blue-roof case has outperformed the base case ($RMSE_{blue-roof\ case} - RMSE_{base\ case} < 0$), while the "RMSE impact" means that the blue-roof case has worsened the CMAQ performance ($RMSE_{blue-roof\ case} - RMSE_{base\ case} \geq 0$). In general, the majority of stations in Guangzhou, Foshan and Dongguan have received a substantial improvement in both January and August, as shown in yellow colour, while some outer stations in southern and eastern parts of PRD and Hong Kong get worse (i.e., RMSE impact) shown in red colour. These stations with the "RMSE impact" designation are primarily suburban areas where a mixed land-use pattern was identified. Overall, stations with "RMSE improvement" yield an average RMSE of 45.8 $\mu g/m^3$ and 30.6 $\mu g/m^3$ for the base and blue-roof cases in January, respectively, which translates to about -12.3 $\mu g/m^3$ for the RMSE improvement. This number is much larger than +0.7 $\mu g/m^3$ in magnitude obtained from the group with the "RMSE impact" designation, which illustrates the improvement has outweighed the impact. For August, the differences in $RMSE_{(blue-roof\ case\ -\ base\ case)}$ under the "RMSE improvement" and "RMSE impact" are -4.5 $\mu g/m^3$ and +0.73 $\mu g/m^3$, respectively. Although there are quite a number of stations (~25+) is fallen into the category of "RMSE impact", their actual RMSE differences are relatively small (e.g., ~75% of stations with RMSE less than 1 $\mu g/m^3$). Hence, it doesn't cause any concern for the blue-roof method. Detailed statistical results for each station have been incorporated into Appendix Table S1 and S2, and the corresponding station locations are available in Appendix Figure S3.

### 3.3.2 Performance comparison between the blue-roof case and point/area based btmpUp case

It is essential to evaluate the performance of which the blue-roof case can perform using observations, while it is also interesting to investigate the difference in the performance of the blue-roof allocation method with the local point/area based bottom-up method under a relatively fine-resolution (i.e., 3 km) condition. In general, the CMAQ simulated $PM_{2.5}$ using the blue-roof method (middle panel of Figure 6) has shown a lower spatial spreading than the one using the point/area based btmUp approach (bottom panel of Figure 6). The low spread of $PM_{2.5}$ in the blue-roof case may be attributed to the insufficient separation of existing industrial emissions. As the blue-roof emission approach took the entire industrial emissions and treated them as location-based emissions without assigning any portion of them to area source, lacking the representation of industrial area sources (e.g., fugitives) in the inventory may have resulted in a less spatial spread, as shown in Figure 5b. Moreover, the base unit of industrial emissions in the current approach is "province-level", which is insufficient to distinguish the industrial speciality for different cities or counties within the domain. From the time-series analysis shown in Figure 7a and b, the RMSE

performance of the blue-roof case (blue line) is quite comparable with the point/area based btmUp case (orange line) and observations (yellow dots). This particular example of the blue-roof case (Figure 7b) can even outperform the point/area based btmUp case in predicting $PM_{2.5}$. From Appendix Table S1 and S2, the average RMSE in January (August) for the base, blue-roof and btmUp cases are 44.8 (25.7) $\mu g/m^3$ 33.3 (22.4) $\mu g/m^3$, and 27.8 (18.3) $\mu g/m^3$, respectively. This illustrates the blue-roof case has outperformed the base case, but still is not as good as the local point/area btmUp case. Figure 9 shows the $PM_{2.5}$ performance of different station types (see Appendix Figure S3). As expected, the point/area based btmUp case has the lowest RMSE among the cases for all station types, while there is a clear improvement of RMSE in urban stations in the blue-roof case; Implementing the blue-roof method has eliminated some of the extreme outliers from the base case, forming a much more narrowed RMSE range. In terms of rural and suburban stations, minor RMSE improvements (i.e., mean values) have been observed. It should be aware that the wider RMSE range showed in the blue-roof case (as compared with the base case) for the suburban category in Figure 9a is just a visual illusion. As the maximum RMSE value of the base case in the suburban category has been plotted as an outliner (dot) instead of a regular line in the upper whisker. Hence, the RMSE range (the two-end whiskers) in the blue-roof case is visually taller than the one in the base case. Appendix Figure S4 shows the station (i.e., CN_1352A) that corresponds to the maximum RMSE in the suburban category, and better performance has been obtained from the blue-roof case (blue line). In the station, the RMSE in January (August) for the base and blue-roof cases are 84.4 (36.0) $\mu g/m^3$ and 50.0 (27.5) $\mu g/m^3$, respectively.

### 3.3.3    Performance of other air pollutants

At last, to better understand the overall impacts on local air quality prediction, Table 3 shows the comparison of performance statistics among the base case, blue-roof case and point/area based btmUp case for $PM_{2.5}$, $NO_2$, and $O_3$. In general, all three pollutants received some improvements when switching from the populated based method to the blue-roof allocation method. A more significant improvement of RMSE is observed in $PM_{2.5}$ and $NO_2$, which ranges from 3.3-11.5 $\mu g/m^3$ for $PM_{2.5}$ and 2.3-2.7 ppb for $NO_2$. The result is somewhat expected as industrial sector is the largest contributor of $PM_{2.5}$ and $NO_2$ emissions in the MIX inventory. In terms of MB, slight degradation is observed in $PM_{2.5}$, which may either be caused by the slight underestimation of total $PM_{2.5}$ emission in GBA or insufficient generation of secondary organic $PM_{2.5}$ from CMAQ, which is commonly observed in version 5.02. For $NO_2$, slight improvement is observed, which is resulted from the removal of large overestimations in the city centre of GZ, FZ, and SZ. Among 75 observation stations, on average, 25 stations received an improvement in RMSE for $PM_{2.5}$ and $NO_2$. The largest RMSE improvement is observed in the Foshan area with -151 $\mu g/m^3$ improvement in $PM_{2.5}$ (See Supplement Fig. S2) and 33 ppb in $NO_2$. This result clearly reflects the weakness and limitation of the population-based method for industrial allocation in the fine-resolution grid. In some stations (i.e., 7), higher RMSE (i.e., an average of 1.8 $\mu g/m^3$ for $PM_{2.5}$ and 1.9 ppb for $NO_2$) are observed. For ozone, minor improvement (i.e., 0.2 ppb and 0.6 ppb in RMSE) has been found which may attribute to the improvement in $NO_2$ prediction and consequently affect the NOx titration process in ozone chemistry. As the improvement is at a marginal level, it is concluded the improvement is limited.

When comparing the blue-roof case with the local point/area based btmUp case, a lower RMSE of $PM_{2.5}$ has been observed in the blue-roof case (Table 3). The difference in the RMSE reflects there is still room for improvement in the blue-roof method. From the large negative MB observed in the MIX emission cases on $PM_{2.5}$, one suggestion would be to scale up the sectorial $PM_{2.5}$ totals from the MIX inventory using an inverse modelling approach (e.g., satellite inversion or source apportionment), which may lead to a better initial $PM_{2.5}$ emission for CMAQ modelling. In terms of $NO_2$ and $O_3$, comparable results (i.e., RMSE) are obtained between the blue-roof and point/area based btmUp cases. Although there is slightly higher RMSE (23.9 ppb vs 18.6 ppb in August) on one of the blue-roof cases, in general, they are all fallen within a similar range of values. In terms of MB, the values in the blue-roof case vary across the seasons, with positive MB on $NO_2$ and negative MB on $O_3$ in January, while positive MB on both $NO_2$ and $O_3$ in August. For the point/area based btmUp case, negative MB has been observed in both January and August. Among the seasons, it is noted that reducing $NO_2$ emission in the blue-roof case in

January may improve the MB of both $NO_2$ and $O_3$ as it reduces the $NO_2$ titration effect in the ozone formation process and causes higher ozone. However, since the MB (i.e., 3 to 5 ppb) of $NO_2$ are relatively small (as compared with the MB of $PM_{2.5}$ (-10 to -15 $\mu g/m^3$), no $NO_2$ adjustment is recommended.

## 3.4    Conclusion remarks

In this work, we developed a new method called the "blue-roof allocation method" for assigning industrial emissions
for the gridded air quality simulation. The proposed method not only provides an alternative way of handling Chinese industrial emissions from the existing population-based method but also allows a higher resolution (up to 3 km) can be generated for local air quality study. As the rapid urban redevelopment and mature public transportation network (e.g., metro/train system) were emerging in China, the relationship of proximity between living place and the workplace was slowly diminished. Hence, the traditional method using population density as a spatial proxy for industrial emissions has become
obsolete.

In the blue-roof allocation method, satellite images from zoom level 14 were applied as the basis for blue-roof extraction. An HSV colour detection algorithm was developed and trained to carry out blue-roof identification. The captured blue-roofs were then converted and stored as individual polygons for further process. The sequence of ArcGIS subprocesses was applied to generate spatial surrogate and gridded emissions. The gridded emissions were tested with CMAQ air quality
simulations for January and August of 2015. The results show that large improvements are observed on both $PM_{2.5}$ and $NO_2$ predictions when compared with the traditional method (i.e., population-based method). By using the blue-roof method, not only reduced the emission errors from large metropolitan areas but also effectively captured the scattered industrial areas located in the rural area. The emission allocation using the blue-roof method has decluttered the urban emissions, allowing better spreading across the region. We are confident that the new method is capable of generating high-resolution input (up to
3km) for local air quality modelling and yield reasonable air quality results. Please aware that the assumption of the blue-roof method where larger blue-roof has more emissions may not always be sufficient under different resolutions. Therefore, further increasing the spatial resolution to lower than 3 km (e.g., 1 km) should be performed with cautions. Before the point/area based bottom-up approach with the unit process data is fully available in China, this method will be a useful technique for handling industrial emissions in China.

## 4    Code/Data availability

The simulated output is available from the corresponding author on reasonable request.

## 5    Author Contribution

The main contributions to model simulation and data management were made by YFL, CCC and XZ; to the data analyses by YFL, CCC and XZ; conceptualization by YFL and JSF; writing -original draft preparation by all co-authors; resources by
380 JCF.

## 6    Competing interests

The authors declare that they have no conflict of interest.

## 7 Acknowledgement

This work was supported by the Innovation and Technology Fund of Hong Kong [ITS/099/16FX] and Research Grants Council, University Grants Committee [21300214]. The author would like to acknowledge Willy Ying, Sairy Wang and Stephen Chan for their supports in the research and the Hong Kong Environmental Protection Department for providing emission inputs. The computations were partially performed using research computing facilities offered by Information Technology Services, the University of Hong Kong.

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

# 9    Figures and Tables

**Table 1. Configuration of CMAQ air quality simulation**

| Configuration | Options |
|---|---|
| Model Code | CMAQ Version 5.0.2 |
| Horizontal Grid Mesh | D1-27km/D2-9km/D3-3km |
|  | (D3 - 152 x 110 grids with total area of ~180,000 $km^2$) |
| Vertical Grid Mesh | 26 Layers |
| Grid Interaction | One-way nesting |
| Initial Conditions | GEOS-CHEM global chemistry model for 27 km domain; finer grid domains based on next coarser grid |
| Boundary Conditions | GEOS-CHEM global chemistry model for 27 km domain |
| **Emissions** | |
| Emissions Processing | MIX with a top-down approach |
| Sub-grid-scale Plumes | No PinG |
| **Chemistry** | |
| Gas-Phase Chemistry | CB05 |
| Aerosol Chemistry | AE5/ISORROPIA |
| Secondary Organic Aerosols | SORGAM |
| Cloud Chemistry | RADM |
| N2O5 Reaction Probability | 0.01 – 0.001 |
| **Horizontal Transport** | |
| Eddy Diffusivity | K-theory |
| **Vertical Transport** | |
| Eddy Diffusivity | ACM2 |
| Deposition Scheme | M3Dry |
| **Numeric** | |
| Gas-Phase Chemistry Solver | EBI |
| Horizontal Advection Scheme | PPM |

**Table 2. Test results of selected locations for the blue roof identification algorithm.**

| Region | Area (Size in $km^2$) | Hit Rate | False Detection Rate | False Alarm Rate |
|---|---|---|---|---|
| Baoding# | Urban (332) | 88% | 51% | 0.5% |
| Shanghai# | Urban (1,336) | 76% | 38% | 0.2% |
| GBA# | Urban (1,194) | 74% | 35% | 0.4% |
| Nagqu | Urban (100) | 92% | 9% | 0.3% |
| Baoji | Suburban (100) | 87% | 54% | 0.2% |
| Kaifeng | Suburban (100) | 76% | 31% | 0.2% |
| Xi'an | Suburban (100) | 76% | 24% | 0.3% |
| Zhengzhou | Suburban (100) | 91% | 8% | 0.1% |
| Taklimakan Desert | Remote (100) | N/A | 0% | 0.0% |
| Yunnan | Remote (100) | N/A | 0% | 0.0% |

# Training areas (only a subarea of its region).

**Table 3. Summary of performance statistics in the case study.**

| Pol. | Mon. | RMSE Base case | RMSE BR case | RMSE Btm Up case | MB Base case | MB BR case | MB Btm Up case | Improvement (RMSE) No. of Station | Improvement (RMSE) BR-Base | Improvement (RMSE) Max. | Worsening (RMSE) No. of Station | Worsening (RMSE) BR - Base | Worsening (RMSE) Max. |
|---|---|---|---|---|---|---|---|---|---|---|---|---|---|
| PM2.5 | JAN | 44.8 | 33.3 | 27.8 | -10.5 | -15.4 | -0.7 | 23 | -22 | -151 | 8 | 2 | 4.6 |
| ($\mu g/m^3$) | AUG | 25.7 | 22.4 | 18.3 | -11.3 | -13.7 | -6.4 | 17 | -8 | -64 | 8 | 1.6 | 2.3 |
| NO2 | JAN | 28.5 | 25.7 | 26.1 | 4.6 | 2.8 | -18.7 | 30 | -5 | -33 | 2 | 1.2 | 1.3 |
| (ppb) | AUG | 26.2 | 23.9 | 18.6 | 5.2 | 3.7 | -11.9 | 28 | -5 | -31 | 11 | 2.6 | 4.1 |
| O3 | JAN | 24.1 | 23.9 | 24.2 | -4.5 | -3.5 | -8.8 | 11 | -2 | -5 | 5 | 1.6 | 1.9 |
| (ppb) | AUG | 31.6 | 31 | 29.9 | 7.2 | 8 | -4.9 | 13 | -2 | -10 | 1 | 1.4 | 1.4 |

Note: Pol: Pollutant; Mon: Month; BR: Blue-roof case; RMSE: Root Mean Square Error; MB: Mean Bias; Max: Maximum. The table on the right only shows the station with ±1 change in RMSE.

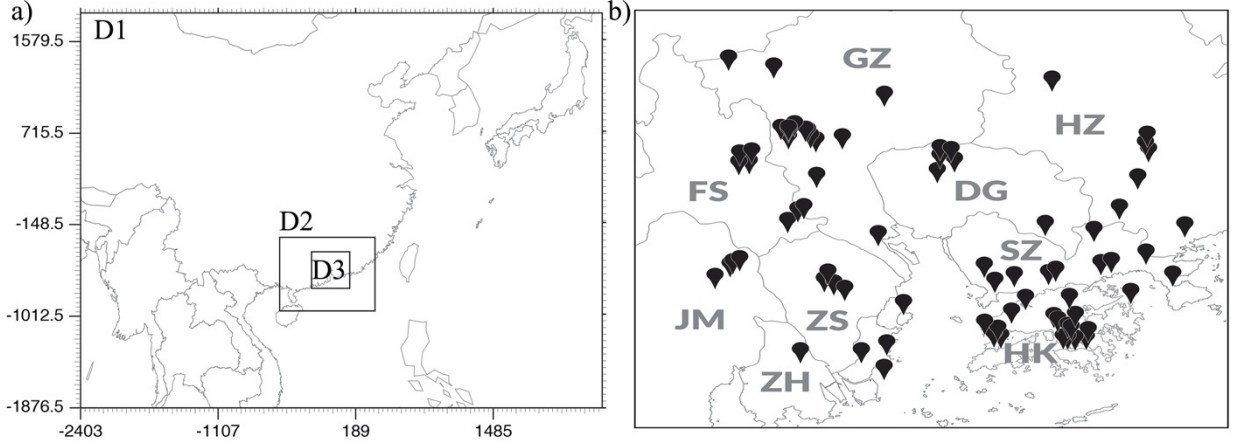

**Figure 1: a) CMAQ simulation domains and b) D3 domain with observations.**

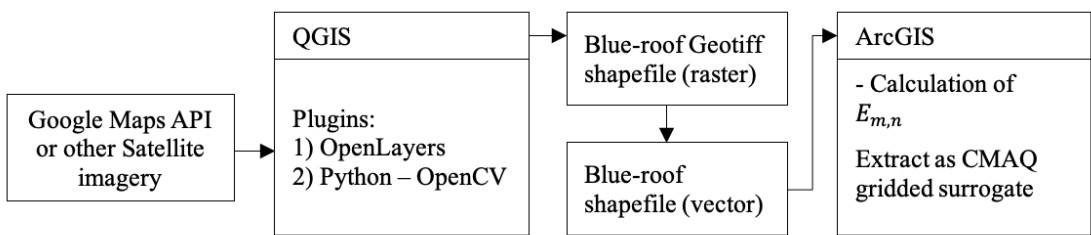

**Figure 2. System flowchart for extract blue-roof industrial buildings.**

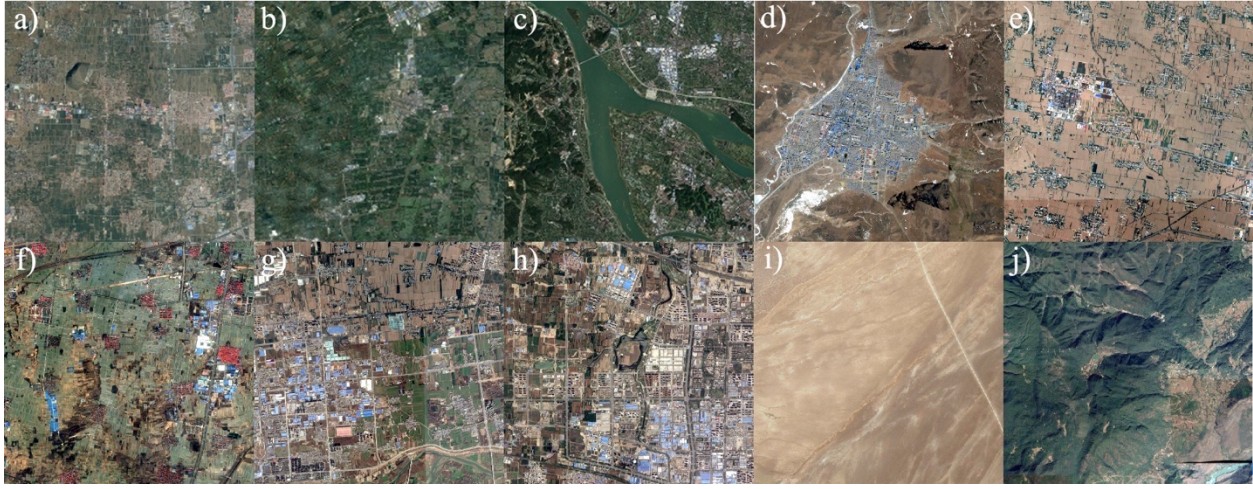

**Figure 3: Selected locations for data training (a-c) and data validation (d-j); a) Baoding, b) Shanghai, and c) GBA, and d) Nagqu, e) Baoji, f) Kaifeng, g) Xi'an, h) Zhengzhou, i) Taklimakan Desert, and j) Yunnan** (Google)**.**

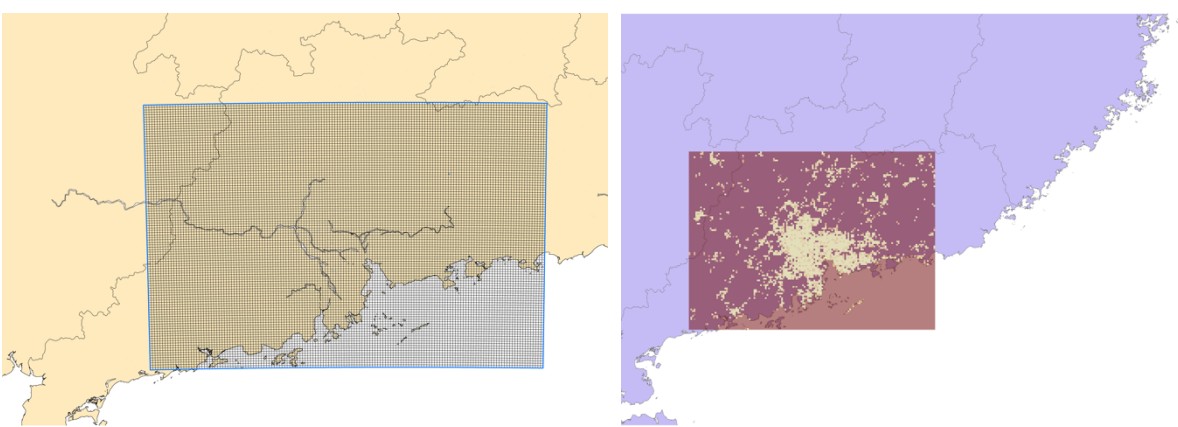

**Figure 4: a) Snapshot of D3 (3km) domain grids, and b) Calculated spatial surrogate.**

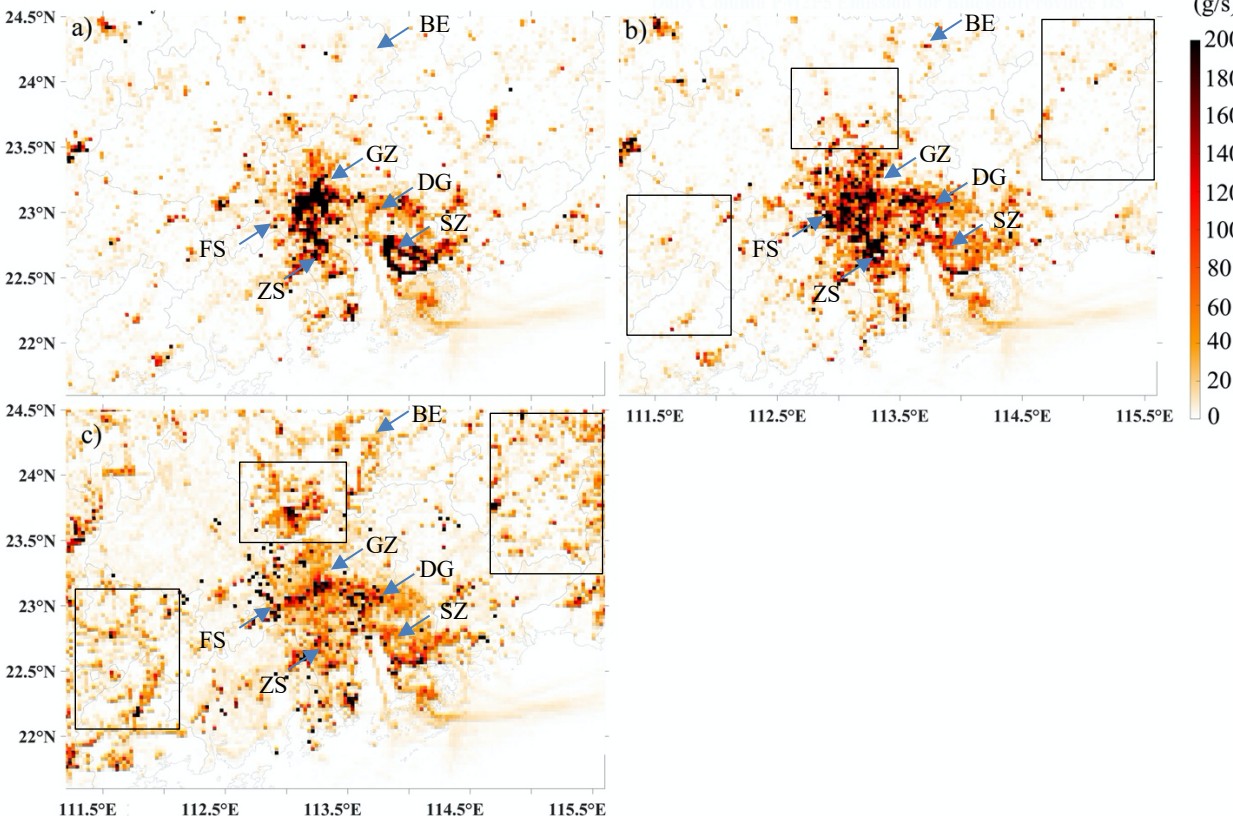

**Figure 5: Daily column total of PM$_{2.5}$ emission from D3 (3 km) domain: a) Base case,  b) Blue-roof case, and c) point/area based BtmUp case. Note: Blue arrows indicate Foshan (FS), Guangzhou (GZ), Shenzhen (SZ), Dongguan (DG), Zhongshan (ZS), and BE (Blue-roof Example). Boxes indicate the locations with large spatial differences between the blue-roof and the btmUp cases.**

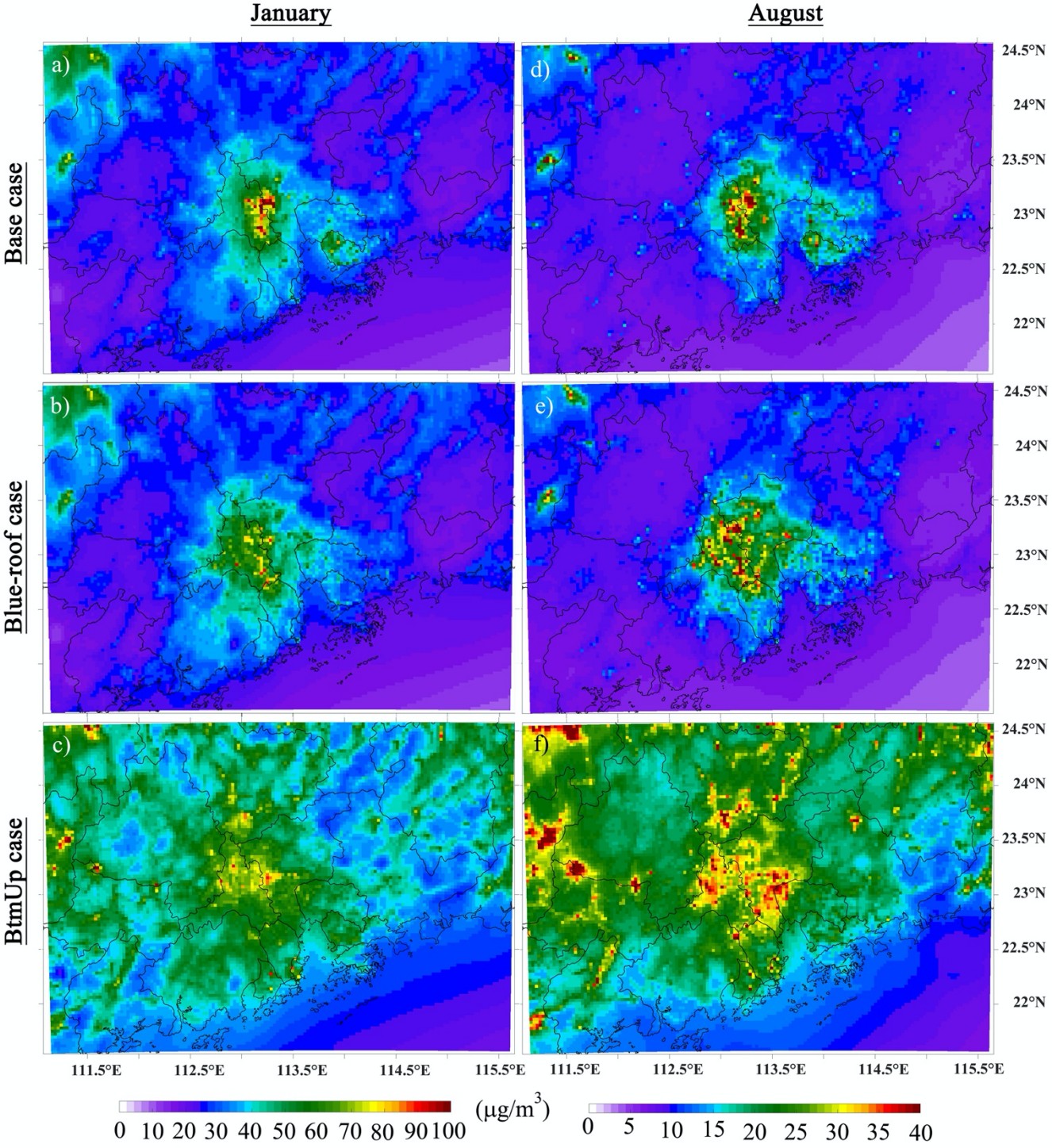

**Figure 6: CMAQ predicted monthly surface PM$_{2.5}$; a) January base case, b) January blue-roof case, c) January point/area based BtmUp case, d) August base case, e) August blue-roof case, and f) August point/area based BtmUp case.**

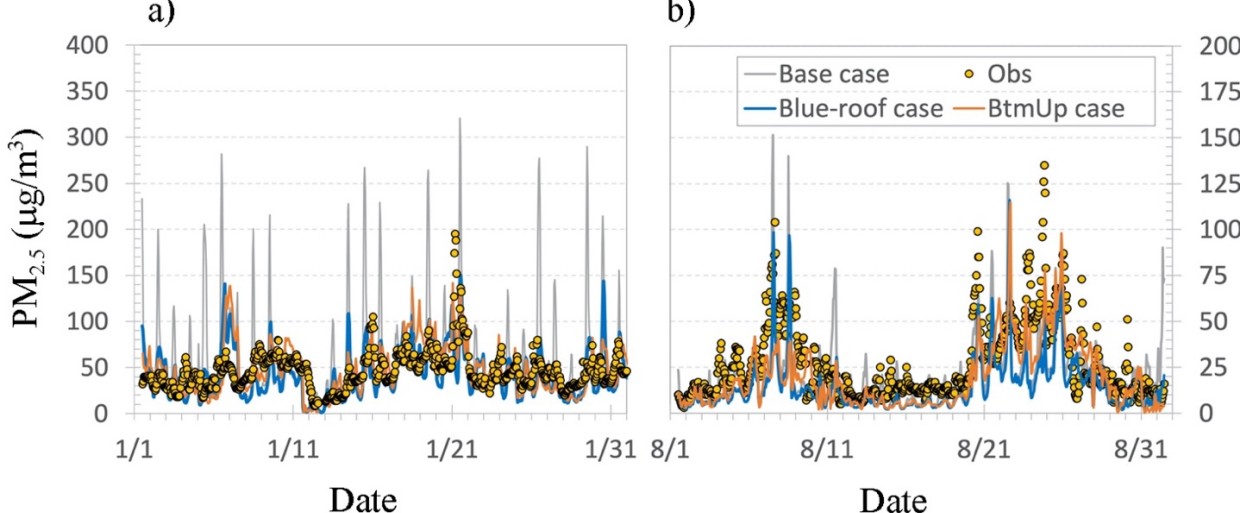

**Figure 7: Time series of PM$_{2.5}$ at station CN_1379A (22°31'16.0"N 113°22'36.8"E) – Zhongshan; a) January and b) August.**

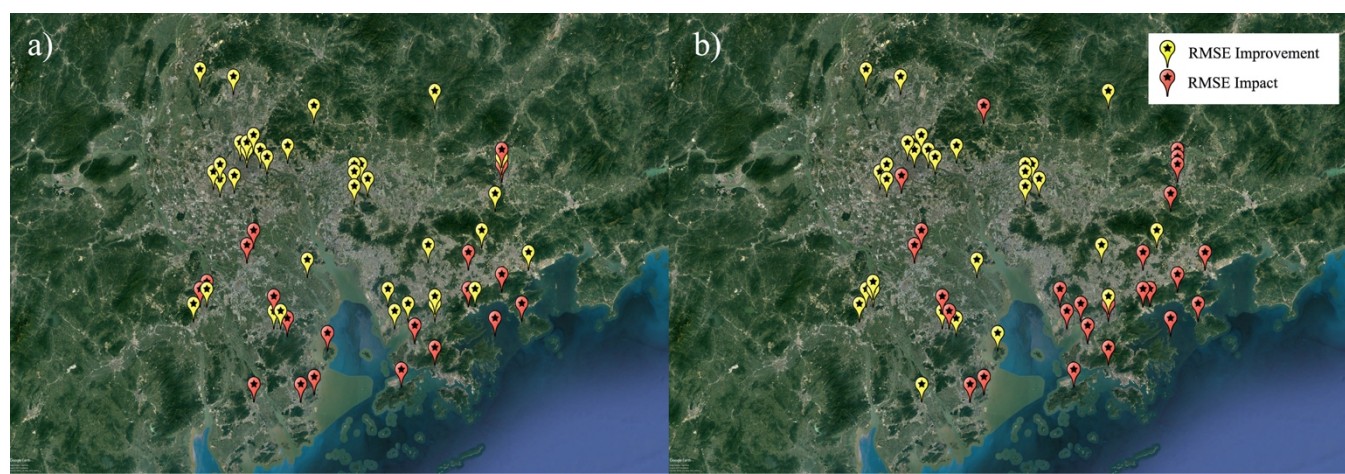

**Figure 8: Spatial comparison of RMSE performance between the base case and blue-roof case: a) January and b) August. Stations with yellow colour indicates "RMSE improvement" where the RMSE of the blue-roof case is lower than the RMSE of the base case (RMSE$_{blue-roof\ case}$ − RMSE$_{base\ case}$ < 0). Stations with red colour refers to as "RMSE impact" (RMSE$_{blue-roof\ case}$ − RMSE$_{base\ case}$ ≥ 0), meaning that the situation gets worse after using the blue-roof algorithm (© (Google)).**

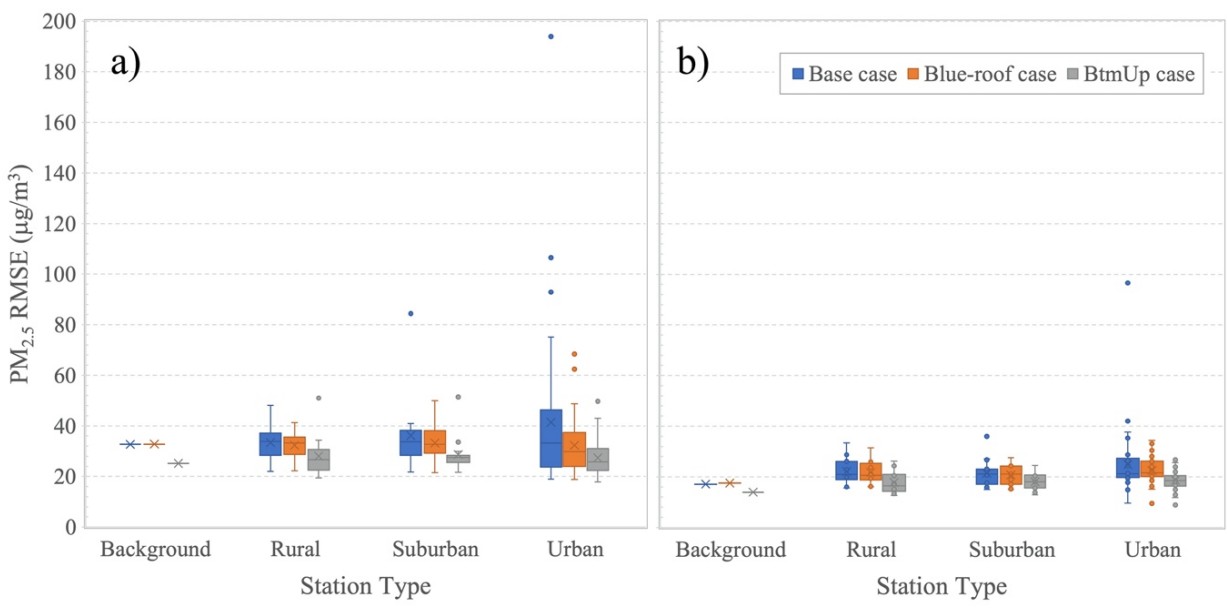

**Figure 9: Performance of PM$_{2.5}$ under different station types: a) January and b) August.**

