# Peer review of "Development of New Emission Reallocation Method for Industrial Source in China"

_Atmospheric Chemistry and Physics, 2020_

## Author Comment (AC1)

Response to Anonymous Referee #1

Light blue: Reviewer comments;
Black: Response to the reviewer;
Black: Add in the manuscript; Black; Underline: Changes in the manuscript

**Comment on acp-2020-1330**

Referee comment on "Development of New Emission Reallocation Method for Industrial Nonpoint Source in China" by Yun Fat Lam et al., Atmos. Chem. Phys. Discuss., https://doi.org/10.5194/acp-2020-1330-RC1, 2021

This paper provided a new industrial nonpoint source reallocation method, a NPS method, based on blue-roof industrial buildings based on satellite imagery. The results indicate using the NPS method will improve the model performance compared with the conventional method of using population density. In general, the paper is clearly organized and easy to follow, and provided a new spatial allocation emission data for the Greater Bay Area (GBA) area. However, the authors mainly compared this method with the regional emission inventories using top-down spatial allocation methods. Currently, the high-resolution emission inventories of most key regions in China adopt point-source processing based on "bottom-up" method. **I encouraged the authors to expand the scope of the paper and to discuss more with the point-based high-resolution emission inventory development.** Some more detailed information should also be provided and discussed, before the paper can be accepted for final publication.

**Response:**

We wish to thank the reviewer for the valuable comments and suggestions, which help us improve the manuscript. For this particular suggestion, we have restructured the paper and added a new emission case (i.e., the point/area based bottom-up case), which utilized a local point/area based inventory in GBA. Detailed changes are covered in the latter of this document.

1) Details follow. As mentioned above, my main concern is that the discussion could be expanded a little bit, and would be focused more on the comparison with point-source based spatial allocation method. That might be more helpful for the whole research community. It invites more review on published work for the high-resolution regional emission inventory, and more comparison and discussion with these inventories I can understand that using the NPS method can effectively improve the spatial allocation of emissions.

**Response:**

Thanks for the suggestion. To address this comment, we have added a new simulation case called the " point/area based bottom-up" (btmUp) case. The bottom-up case is the case that adopts the detailed unit process data and the factory standard industrial Classification (SIC)/Source Classification Codes (SCC)) to derive the emission allocation factors for GBA using the Sparse Matrix Operator Kerner Emissions (SMOKE). The following figure (Figure 5) shows the spatial differences in $PM_{2.5}$ emission among the population based, blue-roof based and point/area based bottom-up approaches. The Figure 5c was added to the existing paper. In order to compare the predictability from these emission methods, additional CMAQ runs using the "bottom-up" case were performed. By comparing the CMAQ results between the blue-roof case and btmUp case, it allows us to better understand the air quality effect from different inventories. Figure 5 shows the spatial comparison of $PM_{2.5}$ emission, and Figure 7 shows the time-series of station CN_1379A. This is the same station which we had used in the last submission. The btmUp case is shown in orange colour. For the detailed discussion, please refer to "Changes in the text" below.

[revised manuscript text omitted]

Note: Pol: Pollutant; Mon: Month; BR: Blue-roof case; RMSE: Root Mean Square Error; MB: Mean Bias; Max: Maximum. The table on the right only shows the station with ±1 change in RMSE.

"When comparing the blue-roof case with the local point/area based btmUp case, a lower RMSE of PM$_{2.5}$ has been observed in the blue-roof case (Table 3). The difference in the RMSE reflects there is still room for improvement in the blue-roof method. From the large negative MB observed in the MIX emission cases on PM$_{2.5}$, one suggestion would be to scale up the sectorial PM$_{2.5}$ totals from the MIX inventory using an inverse modelling approach (e.g., satellite inversion or source apportionment), which may lead to a better initial PM$_{2.5}$ emission for CMAQ modelling. In terms of NO$_2$ and O$_3$, comparable results (i.e., RMSE) are obtained between the blue-roof and point/area based btmUp cases. Although there is slightly higher RMSE (23.9 ppb vs 18.6 ppb in August) on one of the blue-roof cases, in general, they are all fallen within a similar range of values. In terms of MB, the values in the blue-roof case vary across the seasons, with positive MB on NO$_2$ and negative MB on O$_3$ in January, while positive MB on both NO$_2$ and O$_3$ in August. For the point/area based btmUp case, negative MB has been observed in both January and August. Among the seasons, it is noted that reducing NO$_2$ emission in the blue-roof case in January may improve the MB of both NO$_2$ and O$_3$ as it reduces the NO$_2$ titration effect in the ozone formation process and causes higher ozone. However, since the MB (i.e., 3 to 5 ppb) of NO$_2$ are relatively small (as compared with the MB of PM$_{2.5}$ (-10 to -15 $\mu$g/m$^3$), no NO$_2$ adjustment is recommended."

2) However, it seems that the method should not be able to distinguish the differences in emissions between the factories. After all, using images cannot accurately determine the scale of different companies. If it cannot be distinguished, **how much uncertainty was caused by this?**

**Response:**

Thanks for the comments and question. We don't have the actual emissions on any factories. Even for the point/area based bottom-up industry emission inventory, the emission amount is usually based on the industry types instead of an individual company. Also, we can't calculate the uncertainty of the emission estimate from the False Detection Rate and False Alarm Rate in the blue-roof identification from the process. Although the uncertainty estimate of the MIX inventory is given from the literature, that doesn't reflect the uncertainty from our spatial allocation process.

The main target of the study is to show an improvement of applying a blue-roof surrogate for MIX industry sector. The blue-roof algorithm adapts the satellite-image data to produce a more representative emission spatial surrogate without calculating a new emission inventory. The more representative blue-roof surrogate improves the CMAQ model simulation substantially from our analysis. We admit the uncertainty analysis is of great importance in compiling/calculating a bottom-up emission inventory. However, applying the EI uncertainty analysis is not appropriate at the current stage. Instead of showing the EI uncertainty, the developed blue-roof surrogate (Figure 4b), the comparison of the model ready emissions of $PM_{2.5}$, and the time series plots of typical stations are chosen to illustrate the performance of the blue-roof surrogate. More wide-spread emission pattern obtained from Figure5b (applying the blue-roof spatial surrogate) present more consistent pattern with the btmUp case (Figure5c). Figure 7 shows the improvement of the scenario using blue-roof surrogate comparing with the base case. The trend of time series of the Blue-roof case is closer to that of the btmUp case, as expected.

[Figure]

Figure 4: a) Snapshot of D3 (3km) domain grids, and b) Calculated spatial surrogate.

[Figure]

Figure 5: Daily column total of PM$_{2.5}$ emission from D3 (3 km) domain: a) Base case, b) Blue-roof case, and c) point/area based BtmUp case. Note: Blue arrows indicate Foshan (FS), Guangzhou (GZ), Shenzhen (SZ), Dongguan (DG), Zhongshan (ZS), and BE (Blue-roof Example). Boxes indicate the locations with large spatial differences between the blue-roof and the btmUp cases.

[Figure]

Figure 7: Time series of PM$_{2.5}$ at station CN_1379A (22°31'16.0"N 113°22'36.8"E) – Zhongshan; a) January and b) August.

**Add/Changes in text**

**Line 247 -253 in the manuscript**

"When comparing the blue-roof case with the point/area based btmUp case (Figure 5c), clear spots of PM$_{2.5}$ underestimation were observed which are shown in the square boxes of Figure 5b pointing at the northeastern and southwestern sides of PRD, and north of Guangzhou. As the focus of the study is to investigate the improvement of the blue-roof surrogate in the MIX industrial sector, rather than the performance differences between the MIX unified emissions and local bottom-up emissions. Therefore, instead of showing the uncertainty of emission inventory which is infeasible here, we have developed spatial blue-roof surrogate (Figure 4b), the comparison of the model-ready emissions (Figure 5), and the time series plots of typical stations (Figure 7) to illustrate the performance of the blue-roof algorithm."

3) The advantage of this method is to improve the spatial distribution of emissions. However, this study only selected limited city (like Zhongshan in Figure 7) when verifying the model performance. I suggest more monitoring sites should be included to reflect the advantages of improved spatial distribution.

**Response:**

Thanks for the suggestion. We have added two new sections to reflect the spatial performance of the blue-roof study. The first figure (Figure 8) reflects the spatial improvement of $PM_{2.5}$, and the second figure (Figure 9) shows the performance of $PM_{2.5}$ under different types of monitoring stations: a) January and b) August. For the detailed discussion, please refer to "Changes in the text" below.

[Figure]

Figure 8: Spatial comparison of RMSE performance between the base case and blue-roof case: a) January and b) August. Stations with yellow colour indicates "RMSE improvement" where the RMSE of the blue-roof case is lower than the RMSE of the base case ($RMSE_{\text{blue-roof case}} - RMSE_{\text{base case}} < 0$). Stations with red colour refers to as "RMSE impact" ($RMSE_{\text{blue-roof case}} - RMSE_{\text{base case}} \geq 0$), meaning that the situation gets worse after using the blue-roof algorithm (© (Google)).

[Figure]

Figure 9: Performance of $PM_{2.5}$ under different station types: a) January and b) August.

**Add/Changes in text**

**Line 282 -296 in the manuscript**

"Figure 8 shows the comparison of spatial performance between the base and blue-roof cases. The "RMSE improvement" means that the blue-roof case has outperformed the base case ($RMSE_{blue-roof\ case} - RMSE_{base\ case}$ < 0), while the "RMSE impact" means that the blue-roof case has worsened the CMAQ performance ($RMSE_{blue-roof\ case} - RMSE_{base\ case} \geq 0$). In general, the majority of stations in Guangzhou, Foshan and Dongguan have received a substantial improvement in both January and August, as shown in yellow colour, while some outer stations in southern and eastern parts of PRD and Hong Kong get worse (i.e., RMSE impact) shown in red colour. These stations with the "RMSE impact" designation are primarily suburban areas where a mixed land-use pattern was identified. Overall, stations with "RMSE improvement" yield an average RMSE of 45.8 $\mu g/m^3$ and 30.6 $\mu g/m^3$ for the base and blue-roof cases in January, respectively, which translates to about -12.3 $\mu g/m^3$ for the RMSE improvement. This number is much larger than +0.7 $\mu g/m^3$ in magnitude obtained from the group with the "RMSE impact" designation, which illustrates the improvement has outweighed the impact. For August, the differences in $RMSE_{(blue-roof\ case\ -\ base\ case)}$ under the "RMSE improvement" and "RMSE impact" are -4.5 $\mu g/m^3$ and +0.73 $\mu g/m^3$, respectively. Although there are quite a number of stations (~25+) is fallen into the category of "RMSE impact", their actual RMSE differences are relatively small (e.g., ~75% of stations with RMSE less than 1 $\mu g/m^3$). Hence, it doesn't cause any concern for the blue-roof method. Detailed statistical results for each station have been incorporated into Appendix Table S1 and S2, and the corresponding station locations are available in Appendix Figure S3."

**Line 308 -318 in the manuscript**

"Figure 9 shows the PM$_{2.5}$ performance of different station types (see Appendix Figure S3). As expected, the point/area based btmUp case has the lowest RMSE among the cases for all station types, while there is a clear improvement of RMSE in urban stations in the blue-roof case; Implementing the blue-roof method has eliminated some of the extreme outliers from the base case, forming a much more narrowed RMSE range. In terms of rural and suburban stations, minor RMSE improvements (i.e., mean values) have been observed. It should be aware that the wider RMSE range showed in the blue-roof case (as compared with the base case) for the suburban category in Figure 9a is just a visual illusion. As the maximum RMSE value of the base case in the suburban category has been plotted as an outliner (dot) instead of a regular line in the upper whisker. Hence, the RMSE range (the two-end whiskers) in the blue-roof case is visually taller than the one in the base case. Appendix Figure S4 shows the station (i.e., CN_1352A) that corresponds to the maximum RMSE in the suburban category, and better performance has been obtained from the blue-roof case (blue line). In the station, the RMSE in January (August) for the base and blue-roof cases are 84.4 (36.0) $\mu g/m^3$ and 50.0 (27.5) $\mu g/m^3$, respectively."

[Figure]

Figure S4: Time series of surface PM$_{2.5}$ at station CN_1352A (23° 8' 26.628"N 113° 15' 57.24") – North of Guangzhou: a) January and b) August.

4) In "3.4 conclusion remarks", the following sentences seem to be unnecessary repetitions of the body content, and it is recommended to delete them.

**Response:**

Thanks for the suggestion. After some internal discussion with the co-authors, we think that even though the information is partial repeated (as the summary), it serves an important wrap-up for the paper. The last paragraph also provides some take-home messages and recommendations. Therefore, we would like to keep it if possible.

**Add/Changes in text**

**Line 368 -364 in the manuscript**

*"The emission allocation using the blue-roof method has decluttered the urban emissions, allowing better spreading across the region. We are confident that the new method is capable of generating high-resolution input (up to 3km) for local air quality modelling and yield reasonable air quality results. Please aware that the assumption of the blue-roof method where larger blue-roof has more emissions may not always be sufficient under different resolutions. Therefore, further increasing the spatial resolution to lower than 3 km (e.g., 1 km) should be performed with cautions. Before the point/area based bottom-up approach with the unit process data is fully available in China, this method will be a useful technique for handling industrial emissions in China."*

5) There are some mistakes in the MS as well. (1) Line 49. "population density wasa applied as" should be "population density was applied as". (2) Line 134. "In thi study" should be "In this study"

**Response:**

Thank you so much for pointing out that. We have made the changes accordingly.

---

## Author Comment (AC2)

Response to Anonymous Referee #2

Light blue: Reviewer comments;
Black: Response to the reviewer;
Black: Changes in the manuscript;

**Comment on acp-2020-1330**

Referee comment on "Development of New Emission Reallocation Method for Industrial Nonpoint Source in China" by Yun Fat Lam et al., Atmos. Chem. Phys. Discuss., https://doi.org/10.5194/acp-2020-1330-RC2, 2021

The authors developed a new method to allocate industrial emissions onto grid cells based on the areas of blue-roof buildings, which are retrieved from the satellite imagery. This new emission distribution method has been applied to the MIX inventory and evaluated through atmospheric chemistry modeling and comparison against surface observations. Overall, I feel that this study provides new insights into high-resolution emissions mapping, which deserves publication. It has been recognized that the population-based allocation method tends to overestimate anthropogenic emissions over urban areas in China, which could be improved using the method developed in this work to identify the location of blue-roof industrial buildings. **My only concern is that the manuscript lacks a detailed description of the blue roof identification algorithm**, which is difficult to understand in its current form. And the evaluation results using surface observations need further **analysis to illustrate the improvement of the spatial distribution patterns of emission inventories.** My comments are as follows.

**Response:**

We wish to thank the reviewer for providing the valuable comments and suggestions. It helps us to improve the paper further.

Q1), 3.1. The description of the emission allocation method should be moved to the method part in Section 2.

**Response:**

Thanks for pointing out that. I think we had used a wrong description for section 3.1. Now, we have renamed section 3.1 to "HSV value selection, data training, and results of blue-roof colour identification". As this section covers the results of the HSV process, we believe it should belong to section 3 results and discussion. Sorry for the confusion.

Q2), After reading this section, I am not very clear how the blue roof identification algorithm works. What do you mean by "incorporated the effect of sun position on colour change under different latitudinal position in the satellite images." (lines 175-176 on page 5)? Please explain "In this study, 4 ranges of HSV were identified for the blue roof identification algorithm. Its HSV ranges were 193-230° for H, 17% to 90% for S and 40% to 100% for V." (lines 179-180 on page 5)? The method description needs to be further improved to make it easier for an audience to understand.

**Response:**

Thanks for the comments. We have written this part.

**Changes in text**

**Line 181 -193 in the manuscript**

"Three urban areas are Jing-Jin-Ji (Baoding area with 332 km$^2$), Yangtze River Delta (Shanghai area with 1,336 km$^2$), and GBA (Fushan area with 1,194 km$^2$) were picked as the training dataset as we recognized that cities and regions might have their own building styles and development patterns, choosing these three regions not only allowed more diverse samples to be included in the training dataset but also incorporated the potential effect of solar incident angles on image colour (i.e., different brightness) under different latitudinal positions and time of satellite passing. To obtain the "ground truth" reference for iterative comparison, manual digitization of blue-roofs using the zoom level 16 data was performed for those three areas. The result of the iterative process shows that not a single set of HSV ranges was sufficient to capture the blue colour variation exhibited in the google images. As there was a broad spectrum of blue colours (e.g., low cyan, cyan blue, low blue) found in the satellite images, four sets of HSV ranges were used for the blue roof identification algorithm, in which each set of HSV ranges were adopted to identify an independent section of "blue colour" from the HSV solid cylinder. It should be noted that as the ranges of HSV values are considered as business confidential information under the project agreement, the exact values are not disclosed here. In general, the applied HSV values were ranged between 193° and 230° for Hue (H), 17% and 90% for Saturation (S), and 40% and 100% for Value (V)."

[Figure]

Image source for illustration : https://commons.wikimedia.org/wiki/File:HSV_color_solid_cylinder.png

Q3), 3.3. The evaluation of the CMAQ simulation only presents the summary of performance statistics that covers all of the surface observation stations. I am curious whether the model performance (e.g., RMSE, MB) is different among urban, rural, and remote background observation stations, which will help understand the improvement of emission distribution patterns over different regions.

**Response:**

Thank you for the suggestion. We have added some new analysis and discussion on the PM$_{2.5}$ performance. Figure 8 shows the spatial pattern of PM$_{2.5}$ performance: a) January and b) August, 2015. For the detailed discussion, please refer to "Changes in the text" below.

[revised manuscript text omitted]

Q4), Uncertainty assessment. Table 2 presents the False Detection Rate and False Alarm Rate in the blue roof identification algorithm. Is it possible to incorporate this information to quantify the uncertainties in the emission allocation processes?

**Response:**

Thanks for the question. We can't calculate the uncertainty of the emission estimate from the False Detection Rate and False Alarm Rate in the blue-roof identification from the process, as these numbers only reflect the domain-wide performance on blue colour detection. This can't be translated to use in the gridded outputs for uncertainty assessment.

**Response:**

In this study, the MIX emission inventory was adapted to the target simulation year of 2015 (Zhang, 2020; Zhang et al. 2020). For PRD, sector-based control technologies were applied to estimate the emission totals in 2015 (Zheng et al. 2018; Li et al. 2019). The national gas monitoring data (http://www.ipe.org.cn/MapPollution/Pollution.aspx?q=3&type=1), ESRI 2015 population data, and OpenStreetMap traffic data together with the top-down method described in Du (2008) were then temporally and spatially interpreted into 27km (D1), 9km (D2), and 3km (D3) resolutions. The model validation of the base year 2015 was extensively discussed in our previous publication (Zhang et al. 2020). Please note that the trends of emissions from 2012 to 2016 were decreasing, in particularly on $PM_{2.5}$, which has been found to have a large decrease.

**Changes in text**

**Line 110 -117 in the manuscript**

"In this study, the MIX inventory was first scaled to the target simulation year of 2015 based on available sector-based control technologies (Li et al., 2019; Zhang, 2020; Zheng et al., 2018). The derived emission totals from each sector, except for the industrial emissions, were then temporally and spatially interpreted into 27km (D1), 9km (D2), and 3km (D3) resolutions using the top-down emission method described in Du (2008). Detailed methodology and validation of the base year 2015 emission inventory were extensively discussed and can be found in our previous publications (Zhang et al., 2021; Zhang et al., 2020). As Hong Kong emissions were not well presented in the MIX inventory due to the limitation of spatial resolution, the bottom-up emissions from the PATH-2016 platform were adopted for Hong Kong emissions."

Q6), Line 108, page 4. "Hong Kong emissions were not presented in the MIX inventory" If I remember it correctly, the MIX inventory includes Hong Kong emissions.

**Response:**

Thanks for pointing out that. Yes, the MIX inventory includes HK emissions, but it wasn't very well presented under its native resolution (~ 27 km). We would like to say "were not well presented".

**Changes in text**

**Line 113 -115 in the manuscript**

As Hong Kong emissions were not well presented in the MIX inventory due to the limitation of spatial resolution, the bottom-up emissions from the PATH-2016 platform were adopted for Hong Kong emissions."

Q7), Lines 289 and 290, page 8. "However, we are also aware that the assumption of the blue-roof method where larger blue-roof has more emissions may not always be sufficient under different resolutions." This looks very interesting, but the manuscript has not analyzed the emission distribution patterns at different spatial resolutions, right? I would like to know how the authors reach such a conclusion.

**Response:**

In this study, we have assumed the area of blue-roof is directly proportional to the amount of emissions being released. In other words, the bigger factory that occupied a larger piece of land (i.e., roof-top) is assumed to produce more emissions than the small factory which occupies a small land. This is not always true. For example, we could have a small factory using very old combustion technology that produces more emissions than the big factory with advanced technology. In the situation of low-resolution (e.g., 27 km or 9 km), the emission aggregation process that produces individual gridded emissions from many blue-roof areas has damped down the effect during the averaging process. However, when the resolution reaches below 3 km (e.g., 1 km), there are not many factories being averaged. Hence, the chance of falsely allocating the emissions will be much higher.

Our original recommendation on 1 km was mainly based on the nest-down ratio of 3 to 1. So, the next logical resolution in our study is 1 km. However, for the general recommendation, we should not be based on our domain setting. Therefore, it is better to change it to "lower than 3 km", which we had already tested the 3 km in this study.

**Changes in text**

**Line 364 -368 in the manuscript**

"Please aware that the assumption of the blue-roof method where larger blue-roof has more emissions may not always be sufficient under different resolutions. Therefore, further increasing the spatial resolution to lower than 3 km (e.g., 1 km) should be performed with cautions. Before the point/area based bottom-up approach with the unit process data is fully available in China, this method will be a useful technique for handling industrial emissions in China."